# Methanogenesis marker 16 metalloprotein is the primary coenzyme M synthase in *Methanosarcina acetivorans*

Grayson L. Chadwick [1]*, Madison C. Williams [1], Katie E. Shalvarjian [2], Dipti D. Nayak [1,2]*

1 Department of Molecular and Cell Biology, University of California, Berkeley, California, United States of America, 2 Department of Plant and Microbial Biology, University of California, Berkeley, California, United States of America

* chadwick@berkeley.edu (GLC); dnayak@berkeley.edu (DDN)

## Abstract

2-mercaptoethanesulfonate (Coenzyme M, CoM) is an organic sulfur-containing cofactor used for hydrocarbon metabolism in archaea and bacteria. In archaea, CoM serves as an alkyl group carrier for enzymes belonging to the alkyl-CoM reductase family, including methyl-CoM reductase, which catalyzes methane formation in methanogens. Two pathways for the biosynthesis of CoM have been identified in methanogenic archaea. The initial steps of these pathways are distinct but the last two reactions, leading up to CoM formation, are universally conserved. The final step is proposed to be mediated by methanogenesis marker metalloprotein 16 (MMP16), a putative sulfurtransferase, that replaces the aldehyde group of sulfoacetaldehyde with a thiol to generate CoM. Based on prior research, assignment of MMP16 as CoM synthase (ComF) is not widely accepted as deletion mutants have been shown to grow without any CoM dependence. Here, we investigate the role of MMP16 in the model methanogen, *Methanosarcina acetivorans.* We show that a mutant lacking MMP16 has a CoM-dependent growth phenotype and a global transcriptomic profile reflective of CoM-starvation. Additionally, the ∆MMP16 mutant is a CoM auxotroph in sulfide-free medium. These data reinforce prior claims that MMP16 is a *bona fide* ComF but point to backup pathway(s) that can conditionally compensate for its absence. We found that L-aspartate semialdehyde sulfurtransferase (L-ASST), catalyzing a sulfurtransferase reaction during homocysteine biosynthesis in methanogens, is potentially involved in genetic compensation of the MMP16 deletion. Even though both L-ASST and MMP16 are members of the COG1900 family, site-directed mutagenesis of conserved cysteine residues implicated in catalysis reveal that the underlying reaction mechanisms may be distinct.

**Data availability statement:** All data are available in the manuscript or the supplementary materials. All sequencing data have been deposited in the Sequencing Reads Archive and can be accessed under the BioProject number PRJNA1245437.

**Funding:** DDN would like to acknowledge funding from the Searle Scholars Program sponsored by the Kinship Foundation, the Beckman Young Investigator Award sponsored by the Arnold and Mabel Beckman Foundation, the Alfred P. Sloan Research Fellowship sponsored by the Sloan Foundation and the Packard Fellowship in Science and Engineering sponsored by the David and Lucille Packard Foundation. DDN is a Chan-Zuckerberg Biohub – San Francisco Investigator. DDN, GLC and KES were also supported by the Simons Early Career Investigator in Marine Microbial Ecology and Evolution Award sponsored by the Simons Foundation. GLC was also supported by the Miller Institute for Basic Research in Science, University of California Berkeley. KES was also supported by the NSF Graduate Research Fellowship Program (Fellow ID: 202299857). MCW was supported by the NIH Genetic Dissection of Cells and Organisms Training Program (GDTP) (T32 GM 132022). The funders had no role in study design, data collection and analysis, decision to publish, or preparation of the manuscript.

**Competing interests:** The authors have declared that no competing interests exist.

## Author summary

Methane is a high energy renewable fuel that is the primary constituent of natural gas and also a potent greenhouse gas. A significant fraction of global methane emissions is generated by the activity of methanogenic archaea. These microorganisms use a cofactor called Coenzyme M (CoM) as a methyl carrier for methane production mediated by the enzyme Methyl-Coenzyme M reductase. Since methane production is essential for energy conservation in methanogens, they need to synthesize or import CoM. Accordingly, most methanogens encode either one of two CoM biosynthesis pathways. Methanogenesis marker 16 metalloprotein (MMP16) is proposed to catalyze the last step of CoM biosynthesis in both pathways however experimental evidence to this effect is lacking. Here we demonstrate that MMP16 is, indeed, the primary CoM synthase (ComF) in the model methanogen, *Methanosarcina acetivorans*.

## Introduction

2-mercaptoethanesulfonate (Coenzyme M, CoM) is the smallest organic cofactor, consisting of just two carbons joining thiol and sulfonate functional groups. CoM is used in the metabolism of alkenes and alkanes, in bacteria and archaea, respectively. The most broadly distributed and ecologically relevant function of CoM is to act as a methyl-carrier for the final step of methanogenesis in archaea [1]. In this role, the thiol moiety of CoM receives a methyl group either from methyl-tetrahydrosarcinopterin, or directly from a methylated growth substrate via substrate-specific methyl-transferase enzymes. Methyl-Coenzyme M Reductase (MCR), an enzyme unique to methane-metabolizing archaea, reduces Methyl-CoM using Coenzyme B (CoB) to generate methane and the heterodisulfide of CoM and CoB (CoM-S-S-CoB). In anaerobic methanotrophic and alkanotrophic archaea, the net flux of substrates through MCR homologs is in the reverse direction, leading to the consumption of methane or other short chain alkanes.

There are presently three known biosynthetic pathways for CoM, two in methanogenic archaea and one in bacteria (Fig 1). The bacterial pathway for CoM biosynthesis has been completely determined in *Xanthobacter autotrophicus* Py2 and is distinct from the two versions found in methanogenic archaea [2]. The initial steps of the two archaeal pathways vary and use either phosphoenolpyruvate [3] or L-phosphoserine [4] as the starting substrate (Fig 1). Both archaeal pathways converge on sulfopyruvate as a common biosynthetic intermediate, which is decarboxylated by ComDE to sulfoacetaldehyde. In the final step, the aldehyde group of sulfoacetaldehyde is likely replaced by a thiol to produce CoM [3].

Almost all enzymes involved in CoM biosynthesis have been characterized, with the notable exception of the last step in methanogenic archaea i.e. the enzymatic conversion of sulfoacetaldehyde to CoM. This reaction is chemically analogous to the conversion of aspartate semialdehyde to homocysteine by L-aspartate semialdehyde sulfurtransferase (L-ASST) during methionine biosynthesis (Fig 1). In

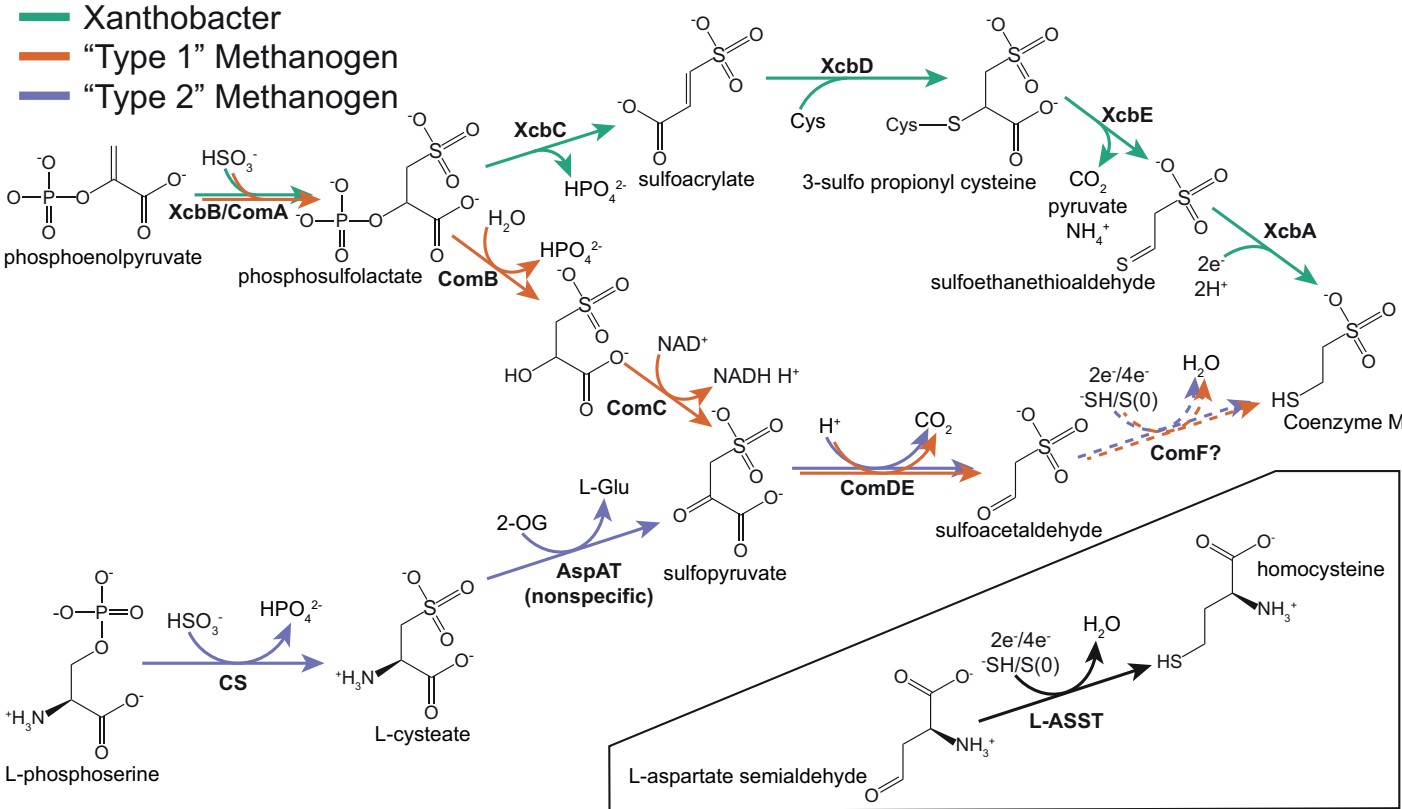

**Fig 1. All known and proposed biosynthetic pathways for 2-mercaptoethanesulfonate (Coenzyme M) in bacteria and archaea.** The bacterial pathway and "Type 1" methanogen pathway begin with phosphoenolpyruvate but diverge after the first step catalyzed by ComA. The "Type 2" methanogen pathway, present in *Methanosarcina acetivorans*, begins with L-phosphoserine and converges with the "Type 1" pathway at sulfoacetaldehyde. The final step of Coenzyme M biosynthesis in methanogens is hypothesized to be carried out by methanogenesis marker protein 16, the putative Coenzyme M synthase (ComF). This reaction is similar to the one catalyzed by L-aspartate semialdehyde sulfurtransferase (L-ASST) (inset), encoded by MA1821-22 in M. *acetivorans*, during homocysteine biosynthesis. The exact nature of the sulfur species utilized by these enzymes is unclear as is the resulting number of electrons required for the reaction. If the sulfur is delivered at the oxidation state of -2 (HS$^-$), the reaction would require an input of 2 electrons whereas if the sulfur is delivered at the oxidation state of 0 (S$^0$), 4 electrons would be required.

the model methanogen, *Methanosarcina acetivorans,* L-ASST comprises of two subunits encoded by MA1821 (MA_RS09480) and MA1822 (MA_RS09485) [5]. MA1821 contains a COG1900 domain thought to be responsible for the aldehyde sulfurtransferase reaction, while MA1822 is a small ferredoxin-containing protein likely involved in electron supply for the reaction [6]. Like L-ASST, methanogenesis marker 16 metalloprotein (MMP16)—a protein family broadly distributed in methanogenic archaea—contains both a COG1900 domain and a ferredoxin domain. This observation led to the initial hypothesis that MMP16 homologs mediates the final step of Coenzyme M biosynthesis in methanogenic archaea [7]. In support of this hypothesis, it was recently reported that *Escherichia coli* can convert sulfoacetaldehyde to CoM when the MMP16 homolog from *Methanocaldococcus jannaschii* (MJ1681) is introduced on a plasmid [8]. Based on this evidence, MMP16 was assigned as the putative Coenzyme M synthase and designated ComF.

However, two independently derived pieces of genetic evidence complicate what would otherwise appear to be a straightforward functional assignment of MMP16 homologs to the final step of CoM biosynthesis. First, a comprehensive transposon mutagenesis screen in *Methanococcus maripaludis* recovered mutants with a disruption in the MMP16 homolog (Mmp1603) on minimal media lacking exogenous CoM [5,9]. Second, it was reported [via personal communication in [8]], that a clean Mmp1603 knockout strain can grow without any CoM-dependence. Since CoM is an integral component

of MCR, which is vital for energy metabolism in methanogens, one would expect the enzymes involved in CoM biosynthesis to also be essential. One possible explanation for this conundrum is genetic compensation by L-ASST i.e. this enzyme can compensate for ComF in its absence [8]. Another possibility is that sulfoacetaldehyde may react slowly with sulfide to produce CoM in an uncatalyzed reaction as has been reported *in vitro* [10]. That said, given the abundance and conservation of MMP16 homologs across the extant diversity of methanogens, it is unlikely that its function is completely redundant to that of another biosynthetic enzyme. Taken together, the role of MMP16 in Coenzyme M biosynthesis, if at all, and its evolutionary connection to the rest of methane metabolism warrants further investigation.

Here we revisit the role of MMP16 in CoM biosynthesis and methanogen physiology. First, through a comparative genomics approach we show that MMP16 co-occurs with other genes involved in CoM biosynthesis across all MCR-containing genomes, supporting its association with CoM biosynthesis through vast evolutionary time. We then show that a mutant of *M. acetivorans* lacking MMP16 can grow, but with a substantial fitness cost, under standard laboratory conditions and is a CoM auxotroph only when exogenous sulfide is eliminated from the growth media. We explored the transcriptional profile of this mutant under various conditions and observed a clear CoM starvation profile, which can be alleviated by the addition of exogenous CoM. Finally, complementation experiments with various MMP16 point mutants improved our understanding of the biochemical function of the COG1900 family. Taken together, our results provide strong evidence that MMP16 (or ComF) is a *bona fide* Coenzyme M synthase however other enzymes of COG1900 family can partially compensate for its absence in the presence of exogenous sulfide.

## Results

### MMP16 homologs are strongly correlated with sulfoacetaldehyde biosynthesis in alkane-metabolizing archaea

We began our investigation into the biological role of MMP16 by exploring its distribution in the genomes of alkane-metabolizing archaea (i.e. those that encode MCR homologs) within the Genome Taxonomy Database (GTDB v. 214.0) [11,12]. If MMP16 is involved in CoM biosynthesis, we reasoned it would only be present in genomes that also have the remainder of the biosynthesis pathway i.e. enzymes that would generate its substrate: sulfoacetaldehyde. To test this hypothesis, we classified all alkane-metabolizing archaeal genomes into four groups, first, based on the presence or absence of sulfoacetaldehyde biosynthesis genes (of either archaeal pathway in Fig 1), and second, based on the presence or absence of MMP16. We found that the presence of MMP16 is strongly correlated with the presence of genes involved in sulfoacetaldehyde production (Fig 2A, p-value = 4e-58, Fisher's exact test). Indeed, several methanogens that lack MMP16 are known CoM auxotrophs, such as *Methanobrevibacter ruminantium* M1[1,13]. Besides this correlation at the whole genome level, we also observed that MMP16 and sulfoacetaldehyde biosynthesis genes physically cluster together in the genomes of many diverse lineages of methane-metabolizing archaea (Fig 2B). This genomic proximity is also suggestive of a shared function.

### Loss of MMP16 leads to CoM dependent growth phenotype in *Methanosarcina acetivorans*

To gain a more complete understanding of CoM biosynthesis—and the role of MMP16 therein—we generated marker-less deletions of MA3297 (MA_RS17200, cysteate synthase; *cs*), MA3298 (MA_RS17205, sulfopyruvate decarboxylase; *comDE*), and MA3299 (MA_RS17210, the putative Coenzyme M synthase; MMP16) in *M. acetivorans*. The second step of the CoM biosynthesis pathway in *M. acetivorans* is catalyzed by a non-specific aminotransferase also involved in amino acid metabolism, so we did not attempt to knock out this step [4]. All mutants were generated using our well-established Cas9-based genome editing system [14] and were validated using whole genome sequencing (S2 Table). We added 1 μM CoM to liquid and agar-solidified medium throughout mutant construction to support the growth of potential CoM auxotrophs. The growth rate of all three mutants were comparable in media supplemented with CoM (Fig 3A). Upon transferring to media without CoM, we observed a dramatic growth defect for the Δ*cs* and Δ*comDE* strains, which is consistent

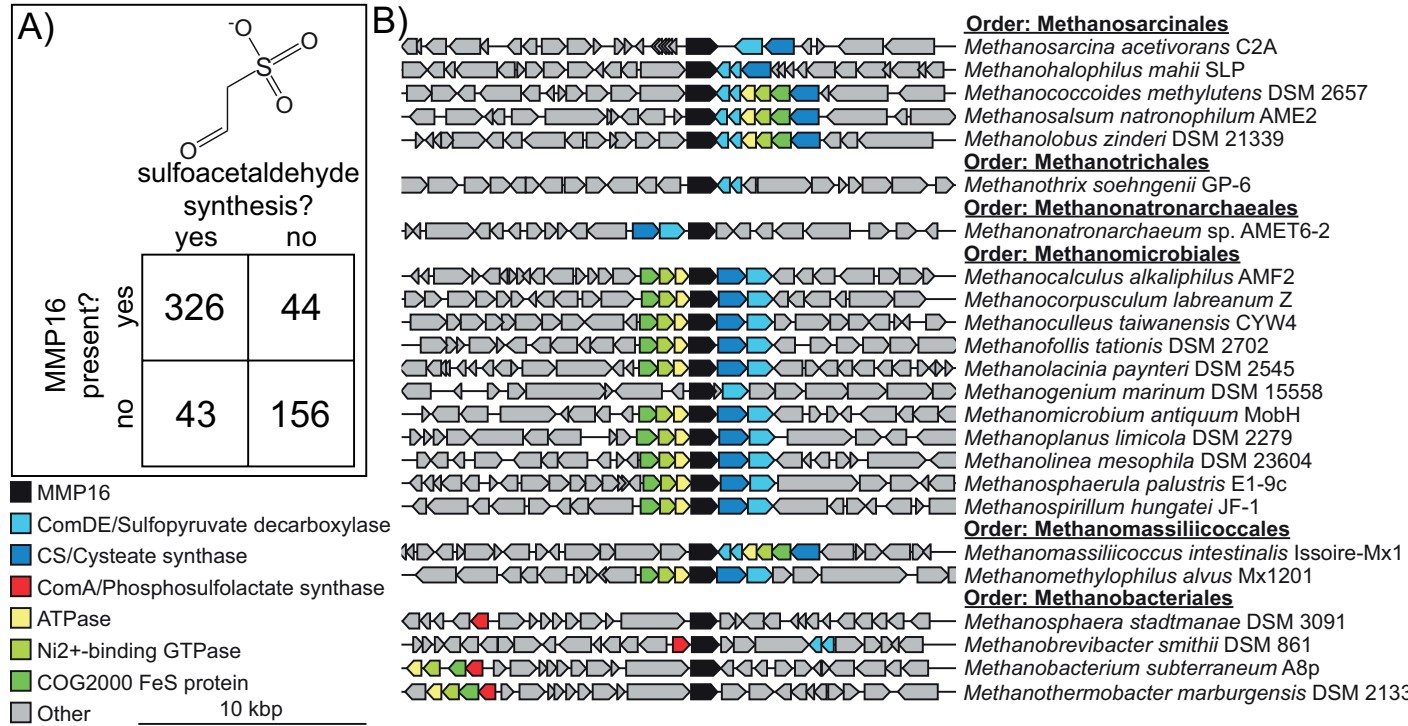

**Fig 2. Comparative genomic analysis of MMP16 and other Coenzyme M (CoM) biosynthesis proteins.** A) The correlation between sulfoacet-aldehyde production and MMP16 presence across all MCR (methyl-coenzyme M reductase)/ACR (alkyl-coenzyme M reductase)-encoding archaeal genomes in the Genome Taxonomy Database (GTDB) (See S1 Table for all genomes). B) Examples of genomes from six different orders of isolated methanogens that display a co-occurrence of MMP16 homologs and other genes involved coenzyme M biosynthesis. Genes encoding ATPase (yellow), a Ni$^{2+}$ binding GTPase (light green) and a COG2000 domain Fe-S protein (dark green) are often colocalized with CoM biosynthesis but their putative role remains unknown. Genetic regions of high completeness sequenced archaeal genomes are displayed in S1 Fig. Note: L-aspartate semialdehyde sulfurtransferase (L-ASST) is not present in many methanogens, and is found in non-MCR containing archaea, supporting the notion that its primary role is not connected with methanogenesis (S1 Table and S1 Fig).

with their known role in CoM biosynthesis (Fig 3A). After a second passage to CoM-deficient media, the Δ*cs* and Δ*comDE* strains behaved like true CoM auxotrophs (Fig 3B). Concentrations of CoM ≥ 1 μM allowed for optimal growth of the Δ*cs* mutant (S2 Fig), hence we used 1 μM as our + CoM condition for the rest of this study. This concentration threshold for CoM is within an order of magnitude of those observed for other natural and artificially generated CoM auxotrophs reported in the literature [15,16].

In contrast to the Δ*cs* and Δ*comDE* strains, the Δ*MMP16* mutant could grow in medium without CoM across multiple passages, albeit at ~70% of the growth rate of the parent strain (Fig 3C). We complemented MMP16 *in trans* on a self-replicating plasmid under the control of the tetracycline inducible promoter P*mcrB(tetO1)*[17]. The *PmcrB(tetO1)* promoter exhibits leaky expression, and previous work has demonstrated that the log$_2$ transformed FPKM value of an uninduced gene driven by this promoter is ~2.2, while induced is ~9.9 [18]. The average wild type expression for MMP16 is ~2.4. Therefore, when complementing ΔMMP16 with MMP16 driven by the *PmcrB(tetO1)*, uninduced is roughly native expression, while induced corresponds to a significant over-expression. Even in the absence of the inducer, growth of ΔMMP16 was restored to wild-type levels in the absence of CoM (Fig 3D). Complementation was specific to MMP16 and not observed for a control protein (Fig 3D). Similarly, the CoM auxotrophy of Δ*cs* is also rescued by complementing

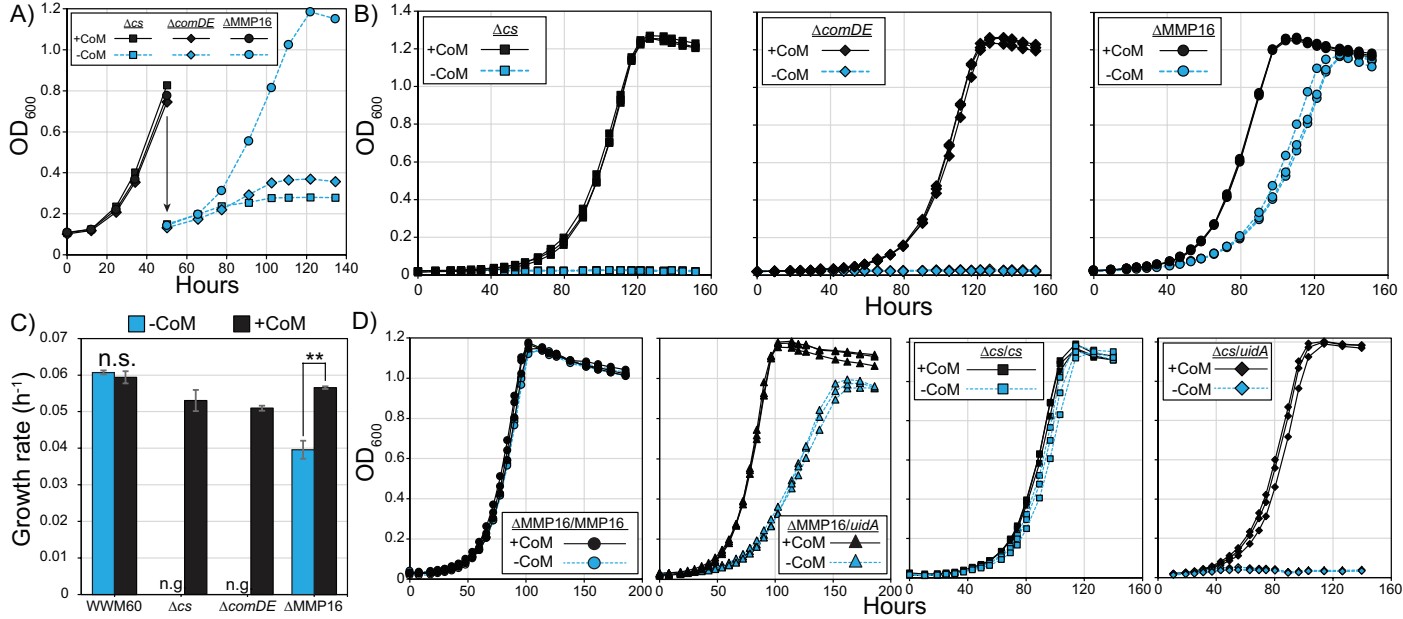

**Fig 3. Growth phenotypes of Coenzyme M (CoM) biosynthesis gene knockouts.** A) Growth of Δ*cs* (MA3297), Δ*comDE* (MA3298) and ΔMMP16 (MA3299) in media containing 1 μM CoM (black), and upon an initial passage into media lacking CoM (blue). The Δ*cs* and Δ*comDE* mutants exhibited a strong growth phenotype in the initial passage unlike the ΔMMP16 mutant. B) Subsequent passaging in triplicate of the CoM-free cultures from panel A results in true CoM auxotrophy phenotypes for Δ*cs* and Δ*comDE* mutants, whereas a significant growth defect is observed for the ΔMMP16 mutant. C) Growth rates calculated from the exponential phase of the parental strain (WWM60) and all three mutants with and without 1 μM CoM (n.s. not significant, n.g. no growth, ** p < 0.01, Student's t-test). Error bars represent standard deviations of three replicate cultures. D) Growth of mutants complemented with the deleted gene or the unrelated *uidA* gene grown in triplicate with and without 1 μM CoM.

the *cs* gene *in trans* (Fig 3D). Importantly, unlike the results reported for *M. maripaludis* [8], we observe a CoM-dependent growth phenotype for MMP16, enabling further investigation into its role in CoM biosynthesis *in vivo*.

## MMP16 is essential in the absence of exogenous sulfide

Our standard media for growth of *M. acetivorans* contains high concentrations of sodium sulfide (0.4 mM), which serves as a reductant as well as a sulfur source for anabolic reactions, including CoM biosynthesis [19]. It has also been reported that sulfide and sulfoacetaldehyde can react abiotically to form CoM under reducing conditions [10]. We hypothesized that the high concentrations of free sulfide in our media might mask the CoM phenotype of the ΔMMP16 mutant in three possible ways: *i)* through an uncatalyzed chemical reaction between sulfide and sulfoacetaldehyde, *ii)* by enabling the promiscuous activity of a non-CoM specific enzyme (such as L-ASST), or *iii)* some combination of the above. Based on this hypothesis, we reasoned that lowering sulfide concentrations in our media may exacerbate the growth defect of our ΔMMP16 mutant in the absence of exogenous CoM.

To test this hypothesis, we grew the ΔMMP16 mutant and the parent strain (WWM60) in media without sulfide. We did not observe any CoM-specific changes in the growth rate of WWM60 across any of the sulfide concentrations tested (Fig 4A). In contrast, we observed a significant growth defect and decreased growth yield for the ΔMMP16 mutant in the absence of sulfide and CoM (Fig 4A), reminiscent of our first passage of the Δ*cs* and Δ*comDE* in media lacking CoM (Fig 3A). Upon an additional passage into sulfide-free medium, the ΔMMP16 mutant behaved like a true auxotroph (Fig 4A). This CoM auxotrophy could be rescued by complementation of MMP16 *in trans* (Fig 4C). Conversely, increasing sulfide to 1.5 mM masked the growth defect of the ΔMMP16 mutant in CoM-free media (Fig 4B). These data may explain why no CoM-dependent growth phenotype was reported for ΔMMP16 in *M. maripaludis* [8], as the standard growth medium for

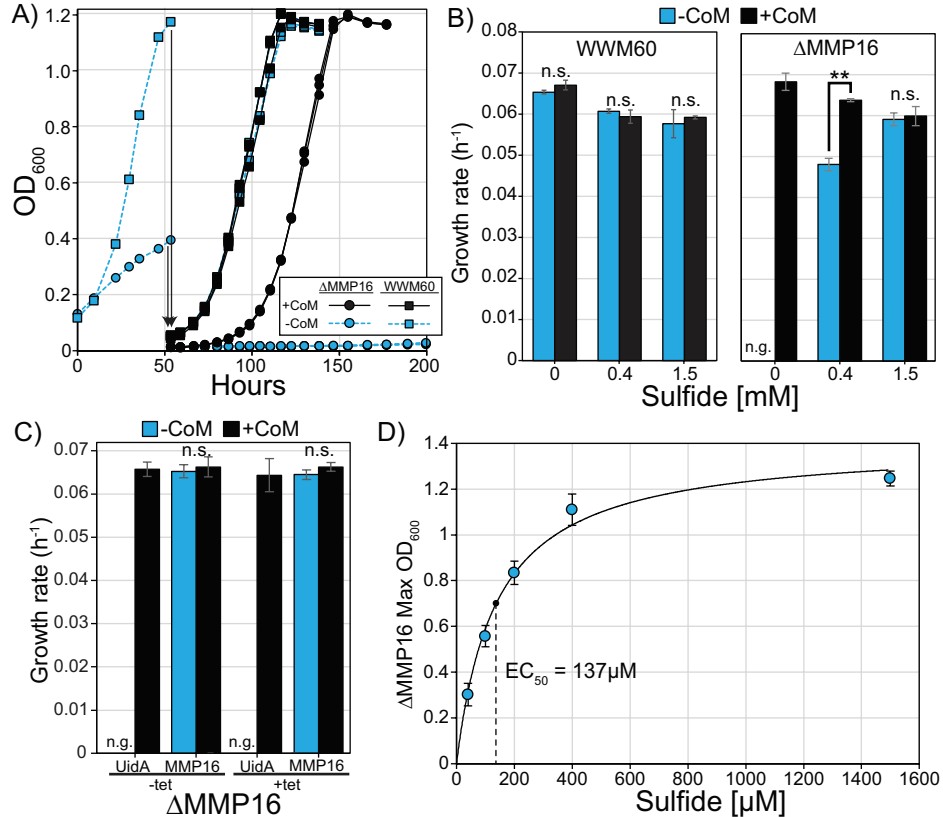

**Fig 4. Growth phenotypes of WWM60 and the ΔMMP16 mutant in media with varying levels of sulfide.** A) Initial growth of WWM60 and the ΔMMP16 mutant in minimal media lacking sulfide and CoM and subsequent passaging in triplicate into sulfide-free media with (black) or without (blue) 1 μM CoM. B) Growth rates of ΔMMP16 and WWM60 as a function of sulfide concentrations. (n.s. not significant, n.g. no growth, ** $p < 0.01$, Student's t-test). C) Complementation of the ΔMMP16 mutant with MMP16 or the unrelated *uidA* gene grown in triplicate with and without 1 μM CoM in media without sulfide and with no tetracycline (-tet) or with 100 μg/mL tetracycline (+tet) to induce expression. D) The maximum optical density of CoM- and sulfide-starved ΔMMP16 upon transfer into media supplemented with varying concentrations of sulfide. Data fit with the Hill equation, yielding an $EC_{50}$ of 137 μM sulfide. Error bars represent standard deviations of three replicate cultures in all panels.

this methanogen contains 2 mM sulfide [20]. Finally, by passaging CoM and sulfide-starved ΔMMP16 into growth media supplemented with a range of sulfide concentrations we observed a clear correlation between exogenous sulfide concentrations (up to 0.4 mM) and growth yield of the ΔMMP16 mutant (Fig 4D).

## The ΔMMP16 mutant has a global transcriptional response to exogenous CoM

The lack of CoM auxotrophy in the ΔMMP16 mutant when grown in the presence of sulfide alludes to the existence of a backup mechanism for the conversion of sulfoacetaldehyde to CoM. To uncover possible mechanisms of genetic compensation in the absence of MMP16, we obtained the transcriptome of the ΔMMP16 mutant and the parent strain (WWM60) in media containing 0.4 mM sulfide either in the presence or absence of 1 μM CoM. Comparison of global transcriptional profile by principal component analysis reveals a strong similarity between WWM60 grown with or without CoM (Fig 5a), which is consistent with the lack of a growth phenotype observed in Fig 3c. In contrast, the transcriptomic profile of ΔMMP16 varied dramatically from the parental strain and exhibited a strong response to the presence of CoM.

The transcriptome of the ΔMMP16 mutant in the absence of CoM had the largest number of differently expressed genes relative to WWM60: nearly 60 genes were differentially expressed (40 up, 20 down, q-value < 0.001 and log2 fold

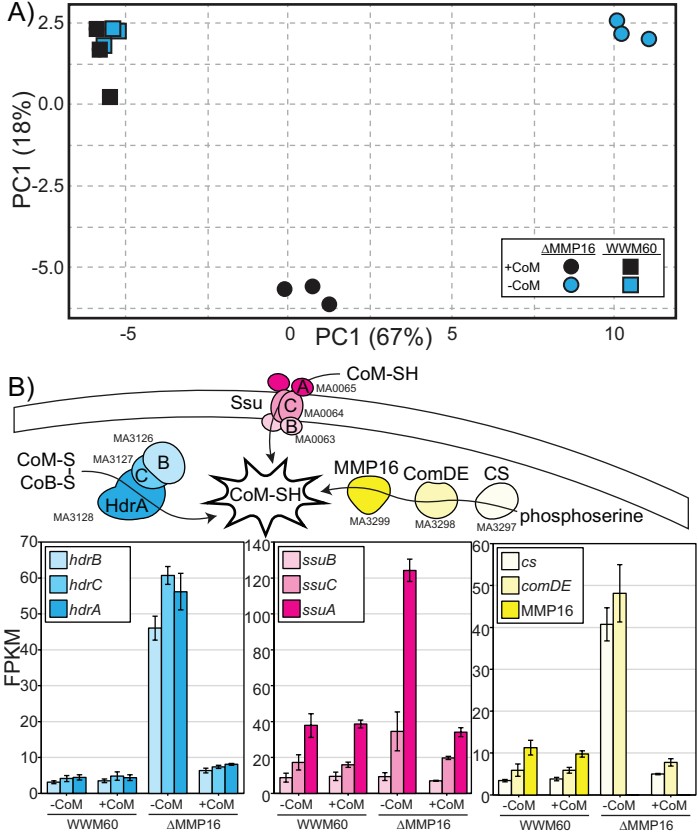

**Fig 5. Transcriptional response of WWM60 and the ΔMMP16 mutant to Coenzyme M (CoM).** A) Principal component analysis of the global transcriptome of ΔMMP16 and the parental strain WWM60 in the presence and absence of CoM. The tight clustering of WWM60 in either growth condition demonstrates its insensitivity to CoM, while a dramatic difference can be observed for the ΔMMP16 mutant between the two treatments. Transcriptomic analyses were conducted with triplicate cultures per strain per treatment. B) A schematic representation of three gene clusters that are among the most differently expressed in the ΔMMP16 mutant shows that they have known roles in CoM biology. Error bars represent standard deviations of three replicate cultures. See S4 Table for all transcriptome data.

change > 1 S3 Table). This list includes two subunits of the CoM-specific ABC transporter (*ssuA* and *ssuC*), the CoM biosynthesis genes *cs* and *comDE*, as well as the CoM-S-S-CoB heterodisulfide reductase *hdrABC* (Fig 5b). We interpret each of these transcriptional changes as a response to CoM starvation. First, by increasing transport of exogenous CoM (futile in this condition), second, by increasing the endogenous biosynthesis of CoM (elevated expression of genes involved in sulfoacetaldehyde production), third, by increasing the recycling of the CoM-S-S-CoB heterodisulfide back to free CoM and CoB. The addition of exogenous CoM to the growth medium for the ΔMMP16 mutant moved the global transcriptome closer to that of WWM60 (Fig 5a) and decreased the total number of genes differentially expressed to 53 (34 up, 19 down). All three pathways upregulated by CoM-starvation described above returned to approximately their wild type expression levels (Fig 5b). The global upregulation of CoM-uptake, biosynthesis and metabolism in the absence of MMP16, that is largely relieved by the supplementation of CoM, further corroborates its role in CoM biosynthesis. In this context, it is interesting to note that a previous study showed that CoM and acetate together were shown to rescue the minor growth defect observed in a ΔhdrABC mutant of *M. acetivorans* [21]. Whether this phenotype is related to the transcriptional response observed here is difficult to assess, as the CoM concentration used in that study is 1 mM, three orders of magnitude greater than the amount of CoM required for optimal growth of a CoM auxotroph.

## L-aspartate semialdehyde sulfurtransferase may compensate for the loss of MMP16

While the transcripts described above reveal a significant CoM-specific response to the loss of MMP16, they do not point to an alternate route for the last step in CoM biosynthesis. The dramatic upregulation of *comDE* and *cs* may result in higher intracellular concentrations of sulfoacetaldehyde, which could improve CoM synthesis through an abiotic reaction or through a promiscuous side-reaction of another enzyme. Given the similarity between MMP16 and L-ASST (MA1821–22), we explored the possibility of L-ASST compensating for MMP16 in its absence.

First, we attempted to knock out L-ASST in WWM60 and the ΔMMP16 mutant. We could readily obtain a ΔL-ASST mutant in the WWM60 background. This mutant grows robustly in minimal media, and its growth rate is unaffected by the addition of either methionine or CoM (S3 Fig). The lack of a methionine-specific phenotype has been demonstrated previously for ΔL-ASST in *M. acetivorans*, and is explained by the presence of a second, orthogonal methionine bio-synthesis pathway [5]. The absence of a CoM-dependent growth phenotype in ΔL-ASST suggests that, unlike MMP16, L-ASST is not the primary pathway for CoM biosynthesis in *M. acetivorans* (S3 Fig). Multiple attempts to delete L-ASST in the ΔMMP16 mutant were unsuccessful, even in media supplemented with CoM or CoM and 3 mM methionine. The conditional essentiality of L-ASST in the ΔMMP16 mutant alludes to a redundant yet essential role of these genes that cannot be rescued by CoM supplementation, likely in CoB biosynthesis [7].

Since we were unable to obtain a Δ L-ASSTΔ MMP16 double mutant, we leveraged the allosteric regulation of L-ASST activity *in vivo* to test its role in CoM biosynthesis (Fig 6A). In *M. acetivorans,* the ferredoxin subunit of L-ASST contains a NIL domain that is known to bind methionine and regulate enzyme activity [22], while the COG1900-containing subunit has a CBS (cystathionine β-synthase) domain that enables allosteric regulation by S-adenosyl methionine (SAM) [23–26]. Thus, we hypothesized that exogenous supplementation of methionine might negatively impact the activity of L-ASST *in vivo* via product inhibition and could be used to test its role in CoM biosynthesis. Indeed, addition of 30mM methionine (in the absence of CoM) exacerbated the growth defect of the ΔMMP16 mutant (Fig 6B). This methionine-induced growth defect was not observed for the ΔMMP16 mutant in the presence of 1 μM CoM.

Despite its putative role in CoM biosynthesis, the expression of L-ASST did not change in the ΔMMP16 mutant. To test if elevated transcription of L-ASST impacts its contribution to CoM biosynthesis, we generated an over-expression mutant in the ΔMMP16 background by introducing these genes *in trans* driven by the P*mcrB*(*tetO1*) promoter. Contrary to our expectations, over-expression of L-ASST imposed a significant fitness cost regardless of the presence of CoM, perhaps due to a dysregulation of methionine metabolism (S4 Fig). Taken together, our data suggest that L-ASST may help compensate for the loss of MMP16, however robust post-transcriptional regulation by methionine and/or SAM may prevent complete recovery of wildtype growth simply by overexpression.

## Conserved cysteine residues in MMP16 are not required for CoM biosynthesis

The COG1900 domains of L-ASST and MMP16 have conserved cysteine residues that are hypothesized to play a critical role in enzyme function[8]. In L-ASST homologs (termed COG1900a) two cysteines are nearly universally conserved: Cys54 and Cys131. While a C54A mutant of L-ASST is non-functional, a C131A mutant is still catalytically active and can mediate methionine production, albeit with diminished activity [5].

Interestingly, in primary sequence alignments, the conserved cysteines in MMP16-type proteins (termed COG1900d) are not in the same position as in COG1900a family members. A previous study speculated that, despite being at different positions in the primary sequence, these conserved cysteines might ultimately reside in similar locations in the three-dimensional enzyme structure [8], allowing them to play an essential catalytic function. To test this hypothesis, we compared the location of well-conserved cysteine residues in MMP16 and L-ASST using Alphafold2 structure predictions. We found that the most conserved cysteines in the COG1900a and COG1900d families were, indeed, located in a similar location in the structure prediction. Cys95 of MMP16 was in a distal loop region similar to the non-essential

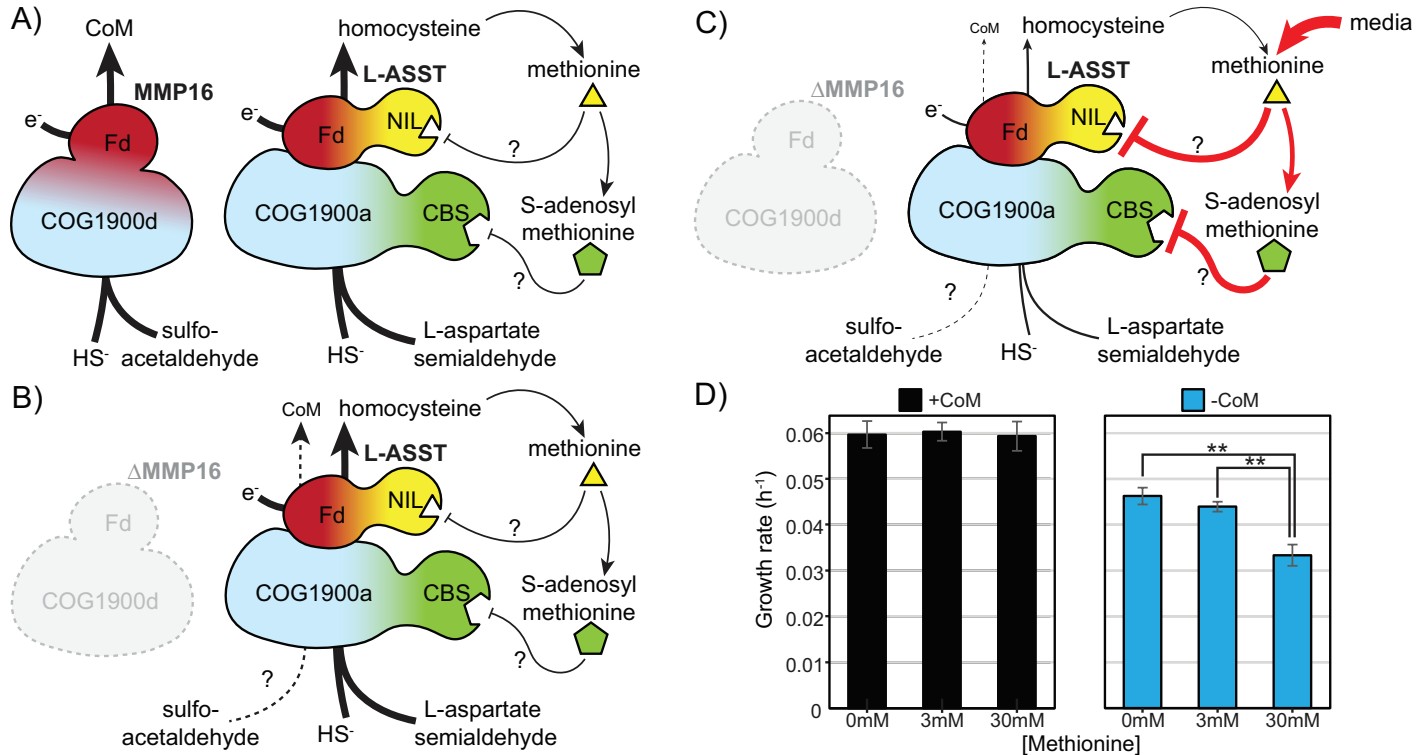

**Fig 6. Methionine supplementation exacerbates the growth phenotype of the ΔMMP16 mutant in the absence of coenzyme M (CoM).** A) Illustration of proposed CoM and homocysteine biosynthesis by MMP16 and L-ASST, respectively, in *M. acetivorans*. The schematic for L-ASST includes the NIL and CBS domains that are hypothesized to undergo product inhibition by methionine and S-adenosyl methionine, respectively. B) In the ΔMMP16 mutant the loss of the primary CoM synthase necessitates a secondary route for CoM production, possibly through L-ASST. C) Exogenous supplementation of methionine could lead to an increase in its intracellular concentration and potentially decrease L-ASST activity post-translationally via the NIL and/or CBS domains. If L-ASST compensates for MMP16 in its absence, this would lead to an incidental reduction in CoM biosynthesis and an exacerbation of the growth defect of ΔMMP16 in the absence of exogenous CoM. D) Growth rates of ΔMMP16 cultures in triplicate in minimal media with varying concentrations of methionine, with and without 1 μM CoM (** $p < 0.01$, Student's t-test). Error bars represent standard deviations of three replicate cultures.

Cys131 in L-ASST, while Cys200/202 were similarly positioned to the essential Cys54 of L-ASST (S5 Fig). Since Cys54 was shown to be essential for L-ASST *in vivo*, we sought to determine if Cys200 and/or Cys202 played similarly essential roles in MMP16.

Unlike the results of the MA1821 mutagenesis experiments, neither C200 nor C202 were found to be necessary for MMP16 activity, either in the presence or absence of sulfide in the growth medium (Fig 7). For some mutants, we observed that the leaky expression of the P*mcrB*(*tetO1*) promoter (i.e. in the absence of the inducer, tetracycline) is insufficient to rescue growth. However, tetracycline-mediated induction of all mutant forms lacking cysteines restored wildtype growth. Thus, it appears that, unlike L-ASST, the conserved cysteine in the core of MMP16 is not essential for its function *in vivo*.

## Discussion

Here we have demonstrated through *in vivo* physiologic and transcriptional investigations that MMP16 is the primary Coenzyme M synthase (ComF) in the model methanogen *M. acetivorans*. The CoM auxotrophy we observed in sulfide-free media and the CoM-specific transcriptional response to the loss of ComF, in combination with a prior report that MMP16 expressed in *E. coli* resulted in the formation of CoM [8], strongly supports this assignment. Further, our

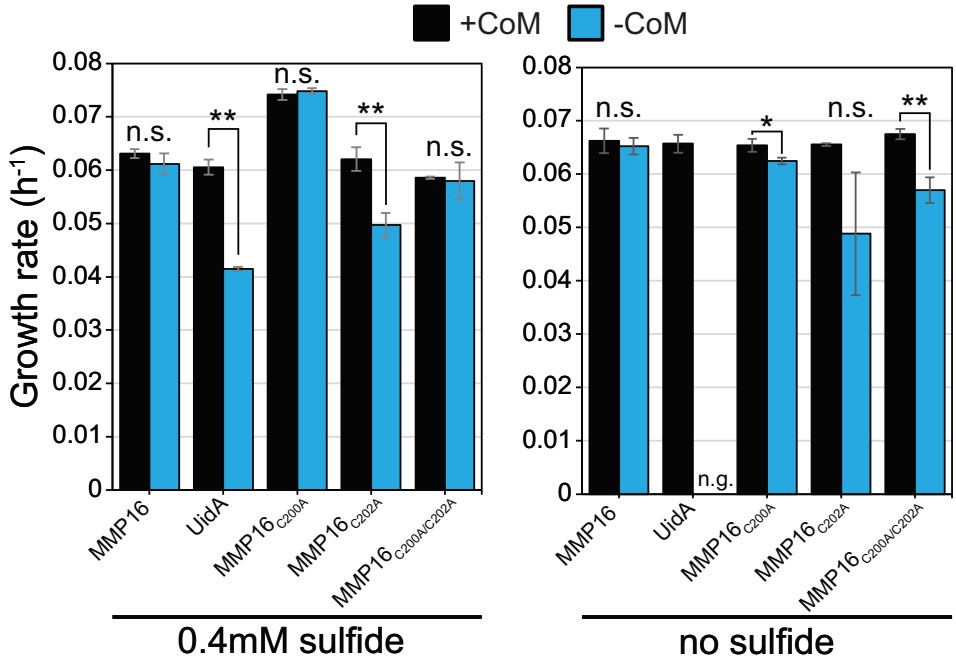

**Fig 7. Growth of the ΔMMP16 mutant complemented with point mutants removing cysteines from the core of the COG1900 domain.**
The ΔMMP16 mutant was complemented *in trans* with genes shown on the X-axis. Cultures were grown in triplicate without tetracycline (or with 100 μg/mL tetracycline; see S6 Fig) in media with 0.4 mM or 0 mM sulfide (n.s. not significant, n.g. no growth, ** p < 0.01, Student's t-test). Error bars represent standard deviations of three replicate cultures.

comparative genomic analysis supports the notion that MMP16 homologs are the main source of CoM produced in methanogenic archaea.

The non-essentiality of MMP16 remains incompletely resolved. At elevated sulfide concentrations (≥0.4 mM), genetic compensation by L-ASST as an alternate coenzyme M synthase seems likely since the addition of methionine, a proposed inhibitor of L-ASST, exacerbates the growth defect of the ΔMMP16 mutant in the absence of CoM (Fig 6). However, we do not have a mechanistic explanation for why the activity of L-ASST as a coenzyme M synthase might decrease as a function of sulfide concentrations. It was found that MA1715 is a sulfur trafficking protein required for the efficient assimilation of sulfide for use by L-ASST [27]. The fact that our *in vivo* point mutant experiments indicated a non-essential role for the conserved Cys202 (Fig 7), and the opposite was found for L-ASST's similarly located Cys54, could imply that these two COG1900 proteins have different interactions with the various sulfur donors such as MA1715, or even different reaction mechanisms entirely. Additionally, our data do not rule out the possibility that a non-orthologous enzyme might be involved or that an un-catalyzed reaction between sulfoacetaldehyde and sulfide in the strongly reducing environment of the methanogen cytoplasm could lead to the formation of CoM. In ongoing work, we are investing the structure and catalytic activity of purified MMP16 and L-ASST *in vitro* free from these possible biotic and abiotic back-up pathways to develop a more complete understanding of the enzymology of the COG1900 family. Regardless of the exact nature of the back-up CoM biosynthesis pathway, our results clearly demonstrate that MMP16 is primary CoM biosynthesis enzyme *in vivo*, and explain the conflicting results in the literature.

Finally, our inability to obtain a ΔMMP16ΔL-ASST double mutant even in the presence of CoM alludes to additional, potentially overlapping, roles for these enzymes beyond the biosynthesis of CoM and homocysteine. Alkane-metabolizing archaea, including methanogens like *M. acetivorans,* produce other redox-active thiols that are important for their biology, notably CoB. The biosynthetic pathways for this compound is still unknown and may require MMP16 and/or L-ASST.

Unraveling the pleotropic role of COG1900 domain proteins in sulfur trafficking for methanogen-specific cofactors would be a promising area for future research.

## Methods

### Plasmid construction

Target sequences for CRISPR-editing plasmids used to delete MA1821–22, MA3297, MA3298 and MA 3299 in *Methanosarcina acetivorans* were designed using the CRISPR site finder tool in Geneious Prime version 2023.0.3 (https://www.geneious.com) as described previously [14] and are listed in S5 Table. The single guide RNA (sgRNA) region with the promoter, scaffold sequence and terminator sequence were amplified using pDN201 as a template with overhangs corresponding to the unique target sequence. The sgRNA was introduced into the Cas9 containing vector pDN201 linearized with *AscI* using Gibson assembly as described before [14]. A homology repair template with a 1000 bp region upstream and downstream of the target locus, to generate an in-frame deletion, was introduced in the sgRNA containing vector linearized with *PmeI* using Gibson assembly as described before [14]. A cointegrate of the CRISPR editing plasmid and pAMG40 was generated using the Gateway BP Clonase II Enzyme mix per the manufacturer's instructions (Thermo Fisher Scientific, Waltham, MA, USA). All plasmids for expression of MMP16 (MA3299) and L-ASST (MA1821–22) *in trans* were generated by introducing the gene(s) in pJK027A linearized with *NdeI* and *HindIII* using Gibson assembly as described previously [14]. A cointegrate of the resulting plasmid and pAMG40 was generated using the Gateway BP Clonase II Enzyme mix per the manufacturer's instructions (Thermo Fisher Scientific, Waltham, MA, USA). Point mutants of MMP16 were generated using primers containing the desired mutation. *E. coli* transformations were conducted using WM4489 and plasmid copy number was induced using 10 mM Rhamnose for purification as described previously [28]. All plasmids were verified by Sanger sequencing at the Barker sequencing facility at University of California, Berkeley. All plasmids and primers used in this study are listed in S5 Table and S6 respectively.

### Mutant generation

A 10 mL culture of *M. acetivorans* in high salt (HS) medium with 50 mM trimethylamine (TMA) in late-exponential phase was used for liposome-mediated transformation with each mutagenic plasmid as described previously [29]. HS medium contains, per liter, NaCl (23.4 g), NaHCO$_3$ (3.8 g), KCl (1.0 g), MgCl$_2$•6H$_2$O (11.0 g), CaCl$_2$•2H$_2$O (0.3 g), NH$_4$Cl (1.0 g), Cysteine•HCl (0.5 g), 1M KH$_2$PO$_4$ (pH = 6.8) (5 mL), 0.2M Na2S•9H$_2$O (2mL), 0.1% resazurin (1 mL), vitamin solution (10 mL), trace elements (10 mL). Vitamin solution contains, per liter, p-Aminobenzoic acid (10 mg), Nicotinic acid (10 mg), Ca panthotenate (10 mg), Pyridoxine HCl (10 mg), Riboflavin (10 mg), Thiamine HCl (10 mg), Biotin (5 mg), Folic Acid (5 mg), α-Lipoic Acid (5 mg), Vitamin B12 (5 mg). Trace elements solution contains, per liter, Nitrilotriacetic acid (trisodium salt) (1.5 g), Fe(NH$_4$)$_2$(SO$_4$)$_2$ (0.8 g), Na$_2$SeO$_3$ (0.2 g), CoCl$_2$•6H$_2$O (0.1 g), MnSO$_4$•H$_2$O (0.1 g), Na$_2$MoO$_4$•2H$_2$O (0.1 g), Na$_2$WO$_4$•2H$_2$O (0.1 g), ZnSO$_4$•7H$_2$O (0.1 g), NiCl$_2$•6H$_2$O (0.1 g), H$_3$BO$_3$ (0.01g), CuSO$_4$•5H$_2$O (0.01 g). Transformants were plated in agar-solidified HS medium with 50 mM TMA, 2 µg/ml puromycin and 1 µM CoM if needed. Plates were incubated in an intra-chamber incubator at 37 ˚C with H$_2$S/CO$_2$/N$_2$ (1000 ppm/20%/balance) in the headspace. Colonies were screened for the desired mutation at the chromosomal locus or the plasmid expressing gene(s) *in trans* and sequence verified by Sanger sequencing at the Barker sequencing facility at University of California, Berkeley (see S6 Table for primers). For gene deletion strains, single colonies that tested positive for the desired mutation were streaked out on HS medium with 50 mM TMA, 20 µg/ml 8ADP (and 1 µM CoM if needed) to cure the mutagenic plasmid. Plasmid cured mutants were verified by screening for the absence of the *pac* gene present on the plasmid with PCR. All strains used in this study are listed in S7 Table.

### Cultivation for Growth Measurements

All *Methanosarcina acetivorans* strains were cultivated in single-cell morphology in hermetically sealed Balch tubes with 10 mL of high-salt (HS) medium supplemented with 50 mM trimethylamine and a headspace of N$_2$/CO$_2$ (80:20) at 8–10

psi as described in [30]. Anaerobic stocks of L-methionine, sodium 2-mercaptoethanesulfonate (Sodium-coenzyme M) and tetracycline hydrochloride were prepared as described previously [17] and added at the desired concentration prior to inoculation. All Balch tubes containing light-sensitive tetracycline were wrapped in aluminum foil to prevent degradation over time. Cultures were incubated at a constant temperature of 37 ˚C in a laboratory incubator (Heratherm series, Thermo Fisher Scientific, Waltham, MA, USA) for growth measurements. Optical density measurements were conducted at 600 nm in a UV-Vis spectrophotometer (Genesys50, Thermo Fisher Scientific, Waltham, MA, USA) outfitted with a holder for test tubes. Doubling times were calculated by performing linear regression of the log2 transformed optical density readings with the highest $R^2$ values. Dose response curves were fit with the Hill equation in the drc R package with a Hill coefficient of 1 to determine the $EC_{50}$ values reported in Figs 4 and S2.

## DNA extraction and sequencing

Genomic DNA was extracted from a 10 mL stationary phase culture of all gene deletion mutants constructed for this study (see S7 Table) using the Qiagen blood and tissue kit per the manufacturer's instructions (Qiagen, Hilden Germany). Library preparation (Illumina DNA Prep kit) and Illumina sequencing (NovaSeq X Plus, 150 bp paired end) was conducted at SeqCenter (Pittsburgh, PA). Demultiplexing, QC, and adapter removal was carried out with bcl-convert (v4.2.4). The sequencing reads were mapped to the *M. acetivorans* C2A reference genome using breseq version 0.38.1 and all mutations in each strain are listed in S2 Table. Raw Illumina sequencing reads are deposited in the Sequencing Reads Archive and and can be accessed under the BioProject number PRJNA1245437.

## RNA extraction, sequencing and transcriptomic analysis

Three replicate 10 mL cultures of WWM60 (parent stain) and DDN290 (the ∆MMP16 mutant) were grown with or without 1 μM CoM at 37 ˚C and 1mL was removed for RNA extraction at an optical density of 0.6. The culture was immediately mixed 1:1 with Trizol (Life Technologies, Carlsbad, CA, USA). After a 5-minute incubation at room temperature the culture and Trizol mixture was applied to a Qiagen RNeasy Mini Kit (Qiagen, Hilden, Germany) and RNA extraction proceeded according to the manufacturer's instructions. DNAse treatment (Invitrogen DNAse (RNAse free)), rRNA depletion (Ribo-Zero Plus kit), cDNA preparation and Illumina library preparation (Stranded Total RNA Prep Ligation) and sequencing (NovaSeq X Plus, 150 bp paired end) were performed at SeqCenter (Pittsburgh, PA). Demultiplexing, QC, and adapter removal was carried out with bcl-convert (v4.2.4). Analysis of transcriptome data was carried out on the KBase bioinformatics platform [31]. Briefly, raw reads were mapped to the *M. acetivorans* WWM60 genome using HISAT2 [32], assembled using StringTie [33], and fold changes, significance values and principal component analyses were calculated with DESeq2 [34]. DESeq2 and StringTie raw data are presented in S4 Table. Raw RNA sequencing reads are deposited in the Sequencing Reads Archive (SRA) and can be accessed under the BioProject number PRJNA1245437.

## Phylogenetic Analyses

MMP16 and COG1900a genes were identified from genomes available in GTDB Release 214.0 [11,12] annotated with Prokka v1.14.5 [35], using gene-specific pHMM's available in the NCBI hmm database. Command line tools developed for automated gene searching with pHMM's, and downstream sequence pulling can be found at the following repository: https://github.com/kshalv/hmm_tools/tree/main. Briefly, this tool iterates through a directory of pHMM's, using HMMER3.4 [36] to search for target genes in a directory of genomes. HMM hits are parsed using SimpleHMMER with an e-value threshold of 1e-03 and organized in an output csv. Hits that exceed the TC score threshold designated in the pHMM are then counted and recorded for each genome in a single output file that is used to generate presence/absence information. Accessions for pHMM's used include: TIGR03269.1 (component A2), TIGR03287.1 (MMP16), and PF01837.20 (COG1900a). Genes for sulfoacetaldehyde biosynthesis (*comA, comB, comC, comDE, comD, comE*, and cysteate

synthase) and MCR catalytic units (*mcrABG*) were identified using the corresponding EC number and custom scripts are documented here: https://github.com/kshalv/coenzymeM.

The tree of methane-metabolizing archaea (S1 Fig) was generated by parsing the GTDB archaeal tree (available in release 214.0) to include genomes with ≥99.0 checkM completeness and all three catalytic subunits of MCR. Genomic neighborhood diagrams for MMP16 and COG1900a were generated using a custom script available in the coenzyme M repository above. The full tree, including genome diagrams and presence/absence information, was visualized using ete3.

### Alphafold structure analysis

Alphafold models of MA3299, MA1821 and MA1822 were retrieved from the AlphaFold Protein Structure Database [37,38]. The MA1821 and MA1822 models were aligned to MA3299 and visualized in Chimera X-1.8 [39]. Alignments of MMP16 and L-ASST proteins were made with Clustal Omega [40].

### Supporting information

**S1 Table. Presence/absence table of CoM biosynthesis genes found in all MCR/ACR containing archaeal genomes (separate file).**
(XLSX)

**S1 Fig. Genomic region surrounding MMP16 and L-ASST across all MCR containing archaea with a CheckM genome completeness score >99% (separate file).**
(PDF)

**S2 Table. Predicted mutations of genome re-sequencing analyzed by Breseq.** The expected deletions were observed in Δ*cs* (MA3297/MA_RS17200), Δ*comDE* (MA3298/MA_RS17205), ΔMMP16 (MA3299/MA_RS17210), and Δ*L-ASST* (MA1821–22/MA_RS09480–85). An additional G->T point mutation was observed in MA_RS02405 in the Δ*comDE* strain. The impact of this additional point mutation was not evaluated.
(DOCX)

**S2 Fig. Growth of Δ*cs* strain relative to CoM concentration in the medium. Above 1 µM no improvement was of the CoM auxotroph was observed, so this concentration was chosen as the concentration for + CoM conditions throughout this work. Data in bottom panel was fit with the Hill equation, yielding an $EC_{50}$ value of 193 nM for CoM. Error bars represent standard deviations of three replicate cultures.**
(AI)

**S3 Table. Protein-coding genes found to be differentially expressed by DESeq2 analysis, using a q-value cutoff of < 0.001 and an absolute value of log2(fold change) > 1.** Total number differentially expressed in black, total number up-regulated (in the row label relative to the column label) shown in blue, total number down-regulated shown in red. All DESeq2 data presented in **Supplementary S4 Table**.
(DOCX)

**S4 Table. DESeq2 and StringTie output from all conditions (separate file).**
(XLSX)

**S3 Fig. Response of *M. acetivorans* to the loss of L-ASST.** As previously reported, *M. acetivorans* is viable without L-ASST as there exists a second pathway for Methionine biosynthesis. Media supplementation with CoM (1 µM) or Methionine (3 mM) did not significantly affect growth rates. The lag observed in cultures with supplemented Methionine is reproducible and the reason behind this lag is unknown. Error bars represent standard deviations of three replicate cultures.
(AI)

**S4 Fig. Effect of over-expressing L-ASST in theΔMMP16 background.** Growth rates from triplicate cultures ofΔMMP16 complemented with either MMP16 itself or L-ASST, induced or uninduced (** $p < 0.01$, Student's t-test). A large growth defect was observed with high over-expression of L-ASST with or without CoM (red lines). This over-expression of L-ASST did not alleviate ΔMMP16's CoM-specific growth defect (black lines). Error bars represent standard deviations of three replicate cultures. (AI)

**S5 Fig. Alignment and Alphafold models of MMP16 and L-ASST.** A) Multiple sequence alignment of MMP16 and L-ASST from four diverse methanogenic archaea highlighting important features. The major domains are highlighted above the sequences and consist of the COG1900 domain, the Ferredoxin domain inserted into the COG1900 domain in MMP16s, and the CBS C-terminal regulatory domain present only in the L-ASSTs. Conserved cysteines making up the inserted Ferredoxin domain are highlighted in gray and indicated with black arrows. The two conserved cysteines in the L-ASST family (Cys54 and Cys131) are boxed in red and indicated with red arrows with *M. acetivorans* numbering. The two conserved cysteines in the MMP16 family (Cys95 and Cys202) are boxed in blue and indicated with blue arrows with *M. acetivorans* numbering. In M. acetivorans there is an additional cysteine (Cys200) close to Cys202, which is not conserved, but which we mutated in order to make sure this cysteine could not compensate for the more conserved, closely positioned Cys202. B) Alphafold models of the MMP16 and L-ASST proteins from *M. acetivorans*. Cysteines in the ferredoxin domains and conserved positions in the COG1900a and COG1900d families are highlighted. Cys54 in L-ASST was found to be essential for *in vivo* function of L-ASST in a prior study, while Cys131 was not, thus we focused on the Cys200 and Cys202 for our mutagenesis studies of MMP16 (Fig 7) due to their similar positioning to Cys54 in the conserved core of the COG1900 domain. (AI)

**S6 Fig. TheΔMMP16 mutant was complemented in trans with genes shown on the X-axis.** Cultures were grown in triplicate without tetracycline (see Fig 7) or with 100 µg/mL tetracycline (below) in media with 0.4 mM or 0 mM sulfide (n.s. not significant, n.g. no growth, ** $p < 0.01$, Student's t-test). Error bars represent standard deviations of three replicate cultures. (AI)

**S5 Table. Plasmids used in this study (separate file).**
(XLSX)

**S6 Table. Primers used in this study (separate file).**
(XLSX)

**S7 Table. Strains used in this study (separate file).**
(XLSX)

**S8 Table. Sequencing statistics for transcriptomic analysis (separate file).**
(XLSX)

**S9 Table. All underlying numerical data for graphs presented in figures (separate file).**
(XLSX)

## Acknowledgments

We would like to members of the Nayak lab for their feedback and input on the manuscript.

## Author contributions

**Conceptualization:** Grayson L. Chadwick, Dipti D. Nayak.

**Data curation:** Grayson L. Chadwick, Madison C. Williams, Katie E. Shalvarjian.

**Formal analysis:** Grayson L. Chadwick, Madison C. Williams, Katie E. Shalvarjian, Dipti D. Nayak.

**Funding acquisition:** Dipti D. Nayak.

**Methodology:** Grayson L. Chadwick, Madison C. Williams, Katie E. Shalvarjian, Dipti D. Nayak.

**Project administration:** Grayson L. Chadwick, Dipti D. Nayak.

**Resources:** Dipti D. Nayak.

**Software:** Katie E. Shalvarjian.

**Supervision:** Grayson L. Chadwick, Dipti D. Nayak.

**Validation:** Grayson L. Chadwick.

**Visualization:** Grayson L. Chadwick, Katie E. Shalvarjian.

**Writing – original draft:** Grayson L. Chadwick, Madison C. Williams, Katie E. Shalvarjian, Dipti D. Nayak.

**Writing – review & editing:** Grayson L. Chadwick, Madison C. Williams, Katie E. Shalvarjian, Dipti D. Nayak.

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
