## [Decision Letter · Decision Letter 0]

2 Apr 2025

PGENETICS-D-25-00234

Methanogenesis marker 16 metalloprotein is the primary coenzyme M synthase in Methanosarcina acetivorans

PLOS Genetics

Dear Dr. Nayak,

Thank you for submitting your manuscript to PLOS Genetics. After careful consideration, we feel that it has merit but does not fully meet PLOS Genetics's publication criteria as it currently stands. Therefore, we invite you to submit a revised version of the manuscript that addresses the points raised during the review process.

Please submit your revised manuscript within 30 days May 02 2025 11:59PM. If you will need more time than this to complete your revisions, please reply to this message or contact the journal office at plosgenetics@plos.org. Please include the following items when submitting your revised manuscript:

We look forward to receiving your revised manuscript.

Kind regards,

Sonja Albers

Guest Editor

PLOS Genetics

Sean Crosson

Section Editor

PLOS Genetics

Aimée Dudley

Editor-in-Chief

PLOS Genetics

Anne Goriely

Editor-in-Chief

PLOS Genetics

**Additional Editor Comments:**

The reviewers found the study very valuable and have included minor corrections to be made in the manuscript. However, it will be critical to describe the sequencing methods more clearly and provide the SRA data for this study.

**Journal Requirements:**

At this stage, the following Authors/Authors require contributions: Madison C. Williams. Please ensure that the full contributions of each author are acknowledged in the "Add/Edit/Remove Authors" section of our submission form.

The list of CRediT author contributions may be found here: https://journals.plos.org/plosgenetics/s/authorship#loc-author-contributions

4) In the online submission form, you indicated that "All sequencing data have been deposited in the Sequencing Reads Archive the bioproject number will be made available upon request. All other data generated in this study will be made available upon request to the corresponding authors.". All PLOS journals now require all data underlying the findings described in their manuscript to be freely available to other researchers, either

- In a public repository

- Within the manuscript itself

- Uploaded as supplementary information.

5) Please amend your detailed Financial Disclosure statement. This is published with the article. It must therefore be completed in full sentences and contain the exact wording you wish to be published. Please ensure that the funders and grant numbers match between the Financial Disclosure field and the Funding Information tab in your submission form. Note that the funders must be provided in the same order in both places as well.

**Reviewers' comments:**

Reviewer's Responses to Questions

**Comments to the Authors:**

Reviewer #1: Interesting contribution to our understanding of coenzyme M biosynthesis in methanogens. The reviewer has only two comments:

Figure 2.B Please check for correctness. e.g. Methanothermobacter marburgensis is not a member of the Methanococcales.

Why ComF/MMP16 rather than MMP16 in Fig. 2A". It would be helpful if in the text to Fig. 2 information with respect to the distribution of the L-aspartate semialdehyde sulfotransferase would also be given. Do all listed archaea contain the corresponding gene(s)?

Authors Summary: Omit last sentence: "Since MMP16 is widely distributed in, and unique to, methanogens it is ideal candidate for the design of anti-methanogen chemical inhibitors". This sentence is misleading for two reasons: (i) with respect to "unique to" MMP16 is also present in methanotrophic archaea and absent in some Methanobrevibacter species; With respect to "ideal candidate", the authors have shown that inhibition (deletion) of MMP16 only leads to a CoM auxotroph in the absence of sulfide, which in anaerobic environments is abundant.

Reviewer #2: The manuscript from Chadwick et al., resolves a long-standing conflict in the methanogenesis field regarding the biosynthetic origin of the essential cofactor coenzyme M (CoM). In particular, prior studies suggest a functional role of the gene production ComF as the enzyme that catalyzed the conversion of sulfoacetaldehyde to CoM in the last step in type 1 methanogens but genetic deletions of the corresponding gene were still viable. In this more detailed studied, the authors show that the comF deletion variant do have a growth phenotype, especially in the absence of exogenous sulfide, and that a housekeeping sulfurtransferase could compensate for the loss of comF, thereby resolving the previously conflicting reports.

The work is nicely carried out and well within the scope of PloS Genetics. My only request is that the authors include the Alphafold2 models that are alluded to in Lines 325-326 as Supplemental Figures as they are important to support their theory that the housekeeping sulfurtransferase may have a distinct mechanism from ComF.

Reviewer #3: The study by Chadwick et al investigates the involvement of the protein MMP16 in the biosynthesis of Coenzyme M, which is an essential cofactor for all methanogenic archaea. While there was prior evidence that MMP16 is involved in this step, there were also contradictory experimental studies that suggested a second, rescue pathway, for the catalysis of this reaction. This present manuscript expands on prior work by showing that MMP16 is a bona fide CoM synthase and elucidates some of the possible rescue pathways.

The authors are commended for the biochemical sleuthing. To me, they could convincingly show that the MMP16 protein is the bona fide CoM synthase, and that high sulfide concentrations in combination with the activity of L-ASST can rescue CoM synthesis. This is rounding up several previous reports in the field with seemingly contradictory results and closes a chapter that has not been concludingly answered in the past. I think this is a beautiful piece of work and I congratulate the authors for their creativity in experimental design and aptitude in conducting them.

I have three main concerns:

L350-380: the discussion is rather short and does not tie together the results obtained by others and the results in this paper; the strength of this paper is that it removes contradictions and is able to explain some of the previously obtained results that seemed unlogical. These are mentioned during the introduction and results but for the sake of clarity I recommend expanding the discussion to include those here.

L382 and following: I find the entire Materials part pretty sparse, a lot “as described previously”. While I understand that it saves time and space for the writer it can be difficult for (inexperienced) readers to find back all these procedures that were described elsewhere. I urge the authors to reconsider this way of describing procedures and rather include all recipes, procedures etc. in the primary text, or in a supplementary material paragraph.

L439, 453: Primary research data are not made available to the reviewer. I think this is not ok. Please let the reviewer see the SRA entry during reviewing.

Minor comments:

L49: it is an ideal candidate

L330: I think the reader could benefit from a display item for this in the main text, e.g. Suppl Fig S6

L374: did the authors try to grow the deltaL-ASST double mutant in the presence of methionine (and CoM)?

L399: with using, remove one

L405: please list HS medium recipe here. I know it was described elsewhere but I think it’s good practice to include as many key procedures as possible.

L410: colonies were screened by PCR? Which primers?

L433: what is a saturated culture? A stationary culture?

L436: there is by far too little information on the sequencing. Which sequencing kit, which platform (Myseq, Hiseq, etc), how many reads per sample, how were samples quality controlled, how were adapters removed etc. Please expand to fully explain the method.

L448: how was the transcriptome collected? Please describe details of the technology and chemistry. Please list statistics of runs. How many reads per sample etc.

**Have all data underlying the figures and results presented in the manuscript been provided?**

Reviewer #1: Yes

Reviewer #2: Yes

Reviewer #3: **No: ** The SRA entries have not been provided.

PLOS authors have the option to publish the peer review history of their article (what does this mean? ). If published, this will include your full peer review and any attached files.

**Do you want your identity to be public for this peer review?** For information about this choice, including consent withdrawal, please see our Privacy Policy .

Reviewer #1: **Yes: ** Rolf Thauer

Reviewer #2: No

Reviewer #3: **Yes: ** Cornelia Welte

**Figure resubmission:**
---

## [Editor Report · Decision Letter 1]

21 Apr 2025

Dear Dr Nayak,

We are pleased to inform you that your manuscript entitled "Methanogenesis marker 16 metalloprotein is the primary coenzyme M synthase in Methanosarcina acetivorans" has been editorially accepted for publication in PLOS Genetics. Congratulations!

Yours sincerely,

Sonja Albers

Guest Editor

PLOS Genetics

Sean Crosson

Section Editor

PLOS Genetics

Aimée Dudley

Editor-in-Chief

PLOS Genetics

Anne Goriely

Editor-in-Chief

PLOS Genetics

Comments from the reviewers (if applicable):

**Data Deposition**

http://datadryad.org/submit?journalID=pgenetics&manu=PGENETICS-D-25-00234R1

**Press Queries**

---

## [Editor Report · Acceptance letter]

PGENETICS-D-25-00234R1

Methanogenesis marker 16 metalloprotein is the primary coenzyme M synthase in Methanosarcina acetivorans

Dear Dr Nayak,

We are pleased to inform you that your manuscript entitled "Methanogenesis marker 16 metalloprotein is the primary coenzyme M synthase in Methanosarcina acetivorans" has been formally accepted for publication in PLOS Genetics! Your manuscript is now with our production department and you will be notified of the publication date in due course.

With kind regards,

Zsofia Freund

PLOS Genetics

On behalf of:
